# Nasopharyngeal swabs vs. saliva sampling for SARS-CoV-2 detection: A cross-sectional survey of acceptability for caregivers and children after experiencing both methods

**François Gagnon**[1]*, **Maala Bhatt**[1,2], **Roger Zemek**[1,2], **Richard J. Webster**[2], **Stephanie Johnson-Obaseki**[3], **Stuart Harman**[1,2]

**1** Division of Pediatric Emergency Medicine, Children's Hospital of Eastern Ontario, Ottawa, Ontario, Canada, **2** Children's Hospital of Eastern Ontario Research Institute, Ottawa, Ontario, Canada, **3** Department of Otolaryngology-Head and Neck Surgery, The Ottawa Hospital, Ottawa, Ontario, Canada

* fgagnon@cheo.on.ca

## Abstract

### Background

Saliva sampling is a promising alternative to nasopharyngeal swabs for SARS-CoV-2 testing, but acceptability data is lacking. We characterize the acceptability of saliva sampling and nasopharyngeal swabs for primary decision makers and their children after experiencing both testing modalities.

### Methods

We administered a cross-sectional survey to participants aged 6-to-17 years and their primary decision makers at an Ottawa community COVID-19 testing centre in March 2021. Included were participants meeting local guidelines for testing. Excluded were those identified prior to participation as having inability to complete the consent, sampling, or survey process. Acceptability in multiple hypothetical scenarios was rated using a 5-point Likert scale. Pain was measured using the Faces Pain Scale—Revised (FPS-R). Preference for testing was assessed with direct binary questions.

### Results

48 participants and 48 primary decision makers completed the survey. Nasopharyngeal swab acceptability differed between scenarios, ranging 79% [95%CI: 66, 88] to 100% [95% CI: 95, 100]; saliva sampling acceptability was similar across scenarios, ranging 92% [95% CI: 82, 97] to 98% [95%CI: 89, 99]. 58% of youth described significant pain with nasopharyngeal swabbing, versus none with saliva sampling. 90% of children prefer saliva sampling. 66% of primary decision makers would prefer nasopharyngeal swabbing if it were 10% more sensitive.

**Data Availability Statement:** All data files are available from the Dryad database at the following DOI: https://doi.org/10.5061/dryad.pnvx0k6qh.

**Funding:** The authors received no specific funding for this work.

**Competing interests:** The authors have declared that no competing interests exist.

**Abbreviations:** FPS-R, Face Pain Scale-Revised; IQR, Interquartile range.

## Conclusion

Though youth prefer saliva sampling over nasopharyngeal swabs, primary decision makers present for testing remain highly accepting of both. Acceptance of nasopharyngeal swabs, however, varies with the testing indication and is influenced by perceived test accuracy. Understanding factors that influence sampling acceptance will inform more successful testing strategies.

## Introduction

Identification and isolation of individuals infected with SARS-CoV-2 are important components of the pandemic control effort [1, 2]. Using nasopharyngeal swabs for SARS-CoV-2 gene detection via reverse transcriptase polymerase chain reaction (RT-PCR) testing is the current criterion standard diagnostic test [3], but recent reviews of saliva sampling have identified this as a potential alternative [3, 4]. Saliva sampling potentially results in less discomfort and risk of viral transmission during testing, decreased utilization of personnel protective equipment, and reduced cost compared to nasopharyngeal swabs [5, 6]. Additionally, nasopharyngeal swabs are contra-indicated in some circumstances (*eg*. nasal anatomic pathology, coagulopathy) and have been associated with rare complications (*eg*. retained swabs, epistaxis, cerebrospinal fluid leakage) [7]. Saliva sampling by expectoration may not be possible for all patients, neonates for example, but saliva can also be sampled with an oral sponge. While pediatric data is under-represented in this literature, the presumption has been that SB sampling would be particularly preferable for children.

In a recent study, school-age children, parents and school personnel were offered COVID-19 surveillance via a saliva sampling method. They were then asked whether they would have consented to testing if a nasal swab was offered instead. Of those who accepted the saliva-based method, 36% would have refused a nasal swab [6]. Of note, more than half of the students in the study had never actually experienced a nasopharyngeal swab when answering the question.

To date there have been no studies comparing perceptions of saliva sampling and nasopharyngeal swabbing amongst children and caregivers who have undergone both sample collection methods. Their opinions could provide valuable insight into the factors affecting willingness to accept COVID-19 testing again in the future. Understanding how acceptable different sampling methods are to those making medical decisions for children, and under what circumstances, would inform future strategies for encouraging appropriate testing. The primary objective of this study was to characterize the acceptability of nasopharyngeal swabs and saliva sampling for COVID-19 in participants who have experienced both modalities. Test acceptability was defined as the willingness of the primary medical decision maker to accept a given test again, in a scenario where testing is indicated. Secondary objectives were to: 1) quantify pain from both saliva sampling and nasopharyngeal swabs; 2) determine which sampling modality was preferred by children; 3) evaluate how a perceived trade-off of test comfort versus accuracy would affect acceptability; and 4) evaluate the interest of primary decision makers for an intranasal topical numbing agent prior to nasopharyngeal swabbing.

## Materials and methods

Approval for this study was granted by the Children's Hospital of Eastern Ontario's ethics review board.

## Study design and setting

A cross-sectional survey was administered to a subset of participants enrolled in a parent study examining the accuracy of saliva testing for SARS-CoV-2 detection. Participants enrolled in the parent study voluntarily presented to an outpatient COVID-19 testing centre (Ottawa, Canada). This sub-study enrolled participants over 3 days in March 2021. Written informed consent was obtained from all participants or their primary decisions makers, and assent was obtained from children whose caregivers were consenting on their behalf.

Children 6 to 17 years old were eligible for inclusion if they: a) were a high-risk contact of a person confirmed to have COVID-19; b) travelled outside of Canada in the last 14 days; and/ or c) had symptoms consistent with COVID-19 infection. Participants were excluded if they: i) did not consent to the parent study (which used both saliva and nasopharyngeal swab testing); ii) had a known developmental delay or intellectual disability that precluded completion of the survey; iii) had nasopharyngeal anomalies, iv) had been previously enrolled in the study; or v) there was a language barrier preventing informed consent.

## Study procedure

Children and caregivers presenting to the regional COVID-19 test centre who consented to the parent study were offered participation in the present sub-study. As part of the parent study, participants underwent both a nasopharyngeal swab and a saliva sample. Nasopharyngeal swabs were inserted into the nasopharynx and gently rotated for 5 to 10 seconds. Saliva samples were collected via 2 different methods based on age. Children 6 to 8 years old had their saliva tested by a sponge-based kit (DNA Genotek; ORE-100). A small sponge was inserted between their cheek and teeth for 1 minute. Children 9 years and older were asked and encouraged to expectorate 1ml of saliva in a tube (DNA Genotek; OM-505). Once both samples were collected by nursing staff, a team member (FG, SH) administered a 15-item questionnaire to the children and their primary decision makers (for medical decisions). Adolescents (12 years old and older) could identify as being their own primary medical decision makers.

As no validated tool to assess our primary outcome existed, we developed a survey according to principles in the Association for Medical Education in Europe (AMEE) guide to questionnaire development [8]. The survey was piloted in families representative of the target population to ensure readability, sensibility and face and content validity. Three hypothetical clinical scenarios were developed to assess acceptability of both tests in different contexts: 1) The study participant requires testing in order to return to school; 2) The study participant has fever and cough and testing is recommended by a health care professional; 3) The study participant is a close contact of someone with COVID-19, but remains asymptomatic. Primary decision makers rated their willingness to accept a test on a 5-point Likert scale ("Very unlikely to Very likely", see Supporting Information S1 File). Procedural pain was assessed using the validated Faces Pain Scale-Revised (FPS-R) [9]. Participants' preference for sampling modality was assessed by asking them to indicate which they would choose if a future test was required, and the desire for an intranasal numbing agent was assessed with a yes or no question. Additionally, primary decision makers were asked to consider a potential trade-off of test comfort versus test sensitivity by indicating their test preference in a hypothetical scenario where saliva sampling was 10% less sensitive than a nasopharyngeal swab.

The survey was administered using an electronic device via secure online database (RED-Cap) [10]. Demographic information was obtained through linkage with the parent study database.

## Statistical methods

To facilitate data analysis, answers to the 5-point Likert were dichotomized to likely and unlikely (including neutral answers). This decision was made during the initial data analysis, with the analyst blinded to any associations with the diagnostic test, as per best practice [11].

Descriptive statistics were calculated for continuous and categorical variables, as medians (with IQR) and frequencies (with proportions), respectively. We compared the proportions of participants answering they were likely to want a test in the 3 clinical scenarios based on which test was offered to them. The estimated proportions and corresponding 95% confidence interval (CI) for pain/preference between clinical scenarios were calculated using the Wilson's score interval. We conducted an exploratory sub-group analysis on age ($<$12 vs. $>$ = 12 years) to better understand the effect of age on the various outcomes. We also carried out an exploratory post-hoc analysis on the effect of the saliva sampling method (expectoration vs. sponge) on outcomes.

We determined a convenience sample size of 50 participants. Data were analyzed with SPSS (version 23; IBM, Armonk, NY).

## Results

### Patient characteristics

Fifty patients were enrolled in the study. Following enrolment, 2 participants were found to be ineligible due to their age. Forty-eight eligible patients were included in the analysis with a median age of 10 years (IQR: 9, 13). Clinical characteristics are presented in Table 1. All

**Table 1. Demographics.**

| | |
|---|---|
| Participants, N | 48 |
| Age, years (Median + [IQR]) | 10 [9,13] |
| Sex, female (N + [%]) | 24 [50%] |
| Saliva sampling method (N + [%]) | |
| *Sponge* | 8 [17%] |
| *Spit* | 40 [83%] |
| NP Swabs before this visit, yes (%) | 23 [48%] |
| # NP Swabs for participants (Median + [IQR]) | 1 [1,3] |
| Primary decision maker NP Swabs, yes (N +[%]) | 27 [57%] |
| # NP swabs for primary decision maker (Median + [IQR]) | 1 [0,2] |
| Symptoms, yes (N + [%]) | 18 [38%] |
| *Fever* | 5 [10%] |
| *Respiratory* | 2 [4%] |
| *Nasopharyngeal* | 9 [19%] |
| *Gastroenterologic* | 11 [23%] |
| *Non-specific* | 11 [23%] |
| Exposure to COVID contact, yes (N + [IQR]) | 31 [65%] |
| Primary decision maker education level (N + [%]) | |
| *Less than high school* | 1 [2%] |
| *High school* | 1 [2%] |
| *College/Trade school/ CEGEP* | 15 [31%] |
| *University* | 31 [65%] |

Abbreviations: N = number, IQR = interquartile range, NP = Nasopharyngeal.

participants enrolled in the study were able to complete both sampling methods and answer the survey.

## Main results

In the scenario in which their child required COVID-19 testing in order to return to school, almost all primary decision makers stated they would be likely to get their child tested independent of the offered sampling modality (98% [95%CI: 89, 99] for saliva method; 100% [95% CI: 95, 100] for nasopharyngeal swab; mean difference 2% [95%CI: -6, 10]; (Fig 1). In the scenario in which their child had a fever and a health care professional recommended testing, most primary decision makers stated they would be likely to get their child tested (98% [95% CI: 89, 99]) for saliva method; 94% [95%CI: 83, 98] for nasopharyngeal swab; mean difference 4% [95%CI: -6, 15]. In the scenario in which their child was in contact with someone who tested positive for COVID, but their child was asymptomatic, more primary decision makers would have their child tested by saliva sampling vs. nasopharyngeal swabs, but the difference was not statistically significant (92% [95%CI: 82, 97] for saliva method; 79% [95%CI: 66, 88]) for nasopharyngeal swab; mean difference 12.5% [95% CI: -1.9, 26].

When the sampling method proposed was a nasopharyngeal swab, participants were more likely to accept testing in the return to school scenario (100% [95%CI: 95, 100]) than the positive contact scenario (79% [95%CI: 66, 88]; mean difference 21% [95%CI: 9, 34]). No significant difference was found between the fever scenario and the other two scenarios for the nasopharyngeal swabs. Participants were equally likely to accept SB sampling for all three scenarios (98% [95%CI: 89, 99] for the return to school scenario; 98% [95%CI: 89, 99] for the fever scenario; 92% [95%CI: 82, 97] for the positive contact scenario).

Fifty-eight percent of children rated their pain to be significant (4/10 or higher) on the FPS-R during the nasopharyngeal swab, whereas no participant described significant pain

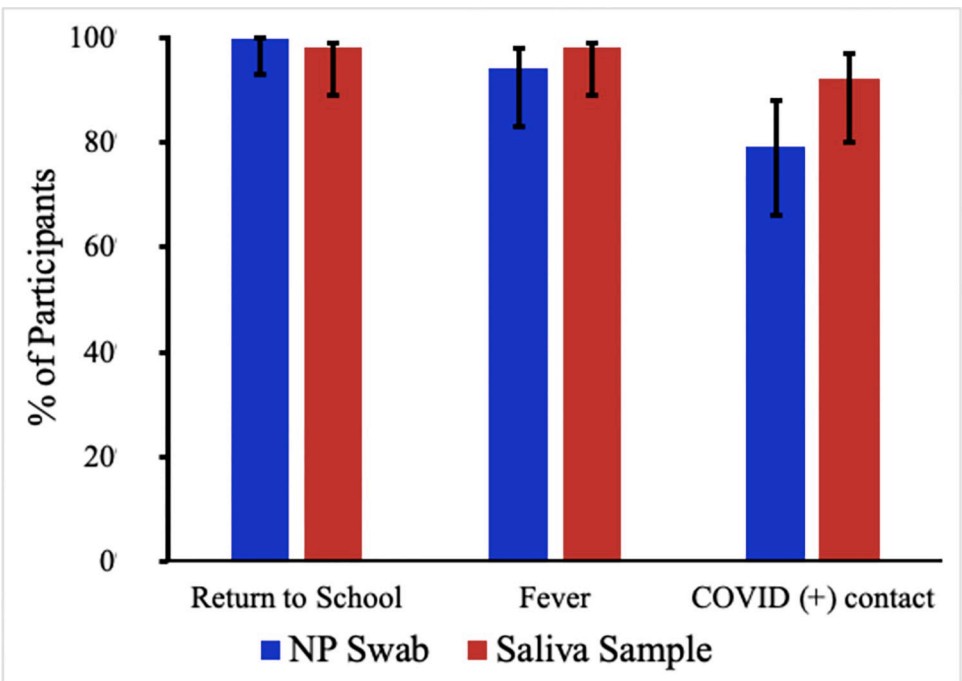

**Fig 1. Percentage of participants responding they would be likely to want COVID-19 testing, based on the sampling method and scenario proposed.**

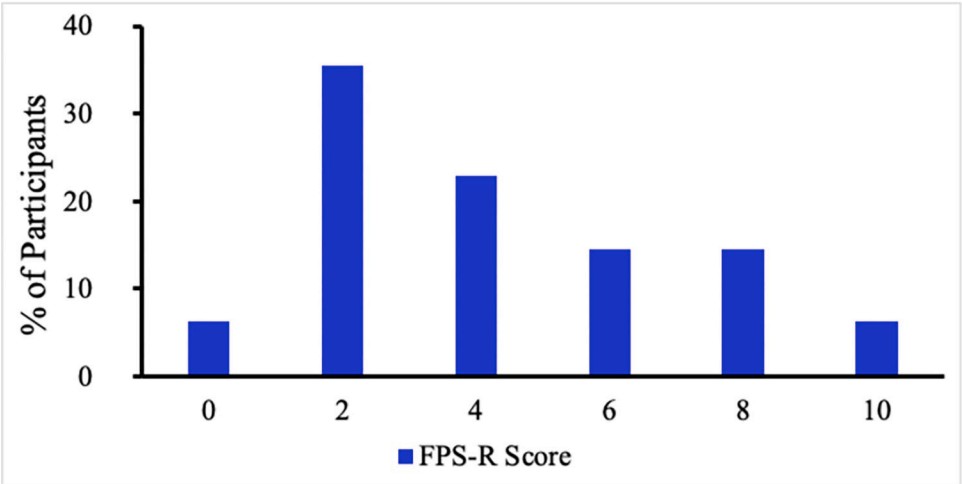

**Fig 2. Self-reported pain after NP swab using revised Face Pain Scale (FPS-R, percent of participants).**

during saliva sampling (Fig 2). Ninety-six percent of children reported 0/10 pain with saliva sampling. Ninety percent of children would prefer to have saliva sampling in the future if they had to be tested again (Fig 3). The two participants who preferred the nasopharyngeal swab stated that it was quicker than the saliva test (expectoration) or that the process of expectorating 1ml of saliva was tedious. All participants (N = 8) who underwent a sponge technique for their saliva collection preferred the saliva test to the nasopharyngeal swab. 66% of primary decision makers indicated they would prefer their child undergo a nasopharyngeal swab if it were true that SB sampling was 10% less sensitive. In this scenario, primary decision makers preferred a nasopharyngeal swab less frequently in younger children (< 12 years) compared to older children (> 12 years) (55% vs 88% respectively). Across all scenarios, the likelihood of

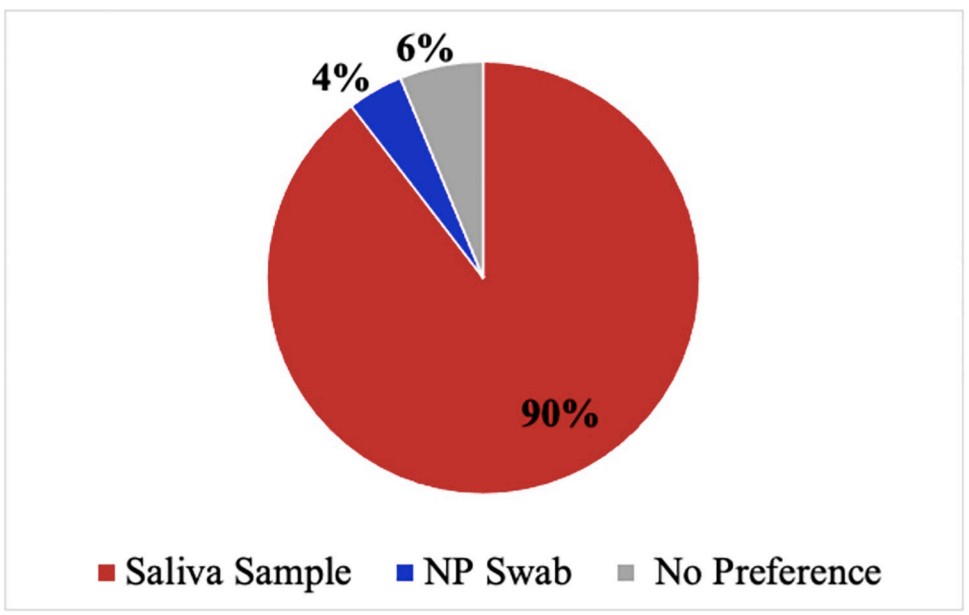

**Fig 3. Preferred test in case of future need for testing (percent of participants).**

accepting a saliva test did not appear to be influenced by the type of saliva testing method used (sponge vs. expectoration).

55% of primary decision makers indicated they would want their child to receive a nasal numbing spray prior to their nasopharyngeal swab even if it meant waiting 10 minutes for the spray to have an effect.

## Discussion

Primary decision makers of pediatric patients 6 years and older who have just experienced a SARS-CoV-2 test are likely to accept testing again, regardless of whether it is a nasopharyngeal swab or saliva sample. Unlike saliva sampling, however, the acceptably of nasopharyngeal swabs was different when the indication for testing changed. Primary decision makers were less accepting of future nasopharyngeal swabs when their child was an asymptomatic contact of a COVID-positive case, compared to when it was a requirement for returning to school. The acceptability of the saliva sample, in contrast, remained high across all scenarios. One of the two participants who preferred a nasopharyngeal swab commented that producing 1ml of saliva was a slow and unpleasant process. It should be noted that most saliva-based assays require only 0.05 to 0.3ml for testing [5]. Alternatively, the sponge collection process appeared to have been well tolerated. After experiencing both tests, the vast majority of children indicated a preference for saliva sampling, which was rated, almost unanimously, as painless. Despite this finding, most primary decision makers indicated they would choose nasopharyngeal swabs over saliva samples if it were true that saliva samples were 10% less sensitive.

Nasopharyngeal swabs do not appear to be unpleasant enough to influence caregiver decisions on future testing. This is similar to the result found by Moisset *et al*. [12], who reported 89% of young adult volunteers in their study rated the acceptability of the nasopharyngeal swab as being 8/10 or higher and 100% would accept the same technique again if indicated. Twenty-eight percent of participants in that study rated pain as being significant (4/10 or higher), which is half the proportion of participants who reported significant pain with nasopharyngeal swabbing in our results. Their trial did not include children, or investigate primary decision maker's willingness to accept a sampling method on behalf of their children.

Primary decision makers in our investigation may have been less accepting of a nasopharyngeal swab in the context of asymptomatic contact with a positive COVID-19 case because the consequence of not testing is less obvious or impactful, compared to the scenario where their child cannot return to school without a test. Given that saliva sampling was highly accepted in all the proposed scenarios, it may be that primary decision makers are more likely to accept this modality over a nasopharyngeal swab when testing is perceived as less valuable (*e.g*., voluntary asymptomatic screening, mild or atypical COVID-19 symptoms). Other studies have shown that COVID-19 test acceptability can vary based on factors of convenience or comfort, such as turnaround time for results [13] or the ability to self-test at home [1]. Further study and delineation of this variability will help determine which sampling options, in which clinical scenarios, most effectively balance test reliability with caregiver willingness to have children tested.

In our study we asked participants if they would accept a 10% reduction in sensitivity using saliva sampling instead of nasopharyngeal swabbing, to determine whether primary decision makers would be willing to trade a reduction in test accuracy for sampling comfort. The actual difference in sensitivity between saliva sampling and nasopharyngeal swabs is not definitively known at present, and likely varies in different clinical scenarios. While initial meta-analyses have estimated sensitivity of saliva samples to be similar to nasopharyngeal swabs [3, 4], Mestdagh *et al*. [14] found that sensitivity varied widely according to saliva collection technique

and viral load. Tan *et al*. [15] also found that test performance varied significantly according to saliva collection and processing technique, and suggested in their review that saliva assays may perform better than nasopharyngeal swabs in optimal conditions.

Determining the ideal test to either screen for, or diagnose, COVID-19 is an on-going area of research that must take into account the circumstances and goals of testing. With the onset of the Omicron variant, which seems to be more transmissible but potentially less severe than previous SARS-CoV-2 variants, testing policies continue to adapt. This will also be true if future variants arise with different clinical characteristics [16]. Test accuracy, cost, feasibility and acceptance are all important factors which must be understood to pivot into effective testing strategies. Our study contributes insight into test acceptability for the pediatric population after children have experienced different sampling methods in a community setting. The implication that acceptability varies by testing rationale more readily when the sampling method is uncomfortable will remain relevant over the evolving course of the pandemic, particularly as the utility and perceived value of testing changes over time. Future studies should further elucidate how different populations respond to the experience of COVID-19 testing in other settings, such as the in-patient setting or when patients are highly symptomatic.

In addition to saliva collection and nasopharyngeal swabs, other alternatives examined in the literature include anterior nasal swabs, throat swabs and combinations of both. It is likely that similar factors identified in this research influence the acceptability of other testing modalities. Additional research is required to further define the acceptability of emerging SARS-CoV-2 testing techniques [5].

## Limitations

There are several limitations to this study. First, as no validated survey to assess SARS-CoV-2 test acceptability exists in the literature, we had to develop a survey to assess our primary outcome, based on best practices [8]. Second, this sample consists of those who went to assessment centers for testing and then consented to be included in a research study, which may affect generalizability. However, the benefit of obtaining the opinions of youth and families who experienced testing with both modalities is valuable regardless of selection bias. The education level of the recruited families was higher than what might be expected of the general population and could have an impact on the generalizability of the results. This is possibly due to the urban location of the testing center.

## Conclusion

Though most youth find saliva sampling painless and prefer it to nasopharyngeal swabs, primary decision makers present for the experience generally remain accepting of both methods for COVID-19 testing. Acceptance of nasopharyngeal swabs, however, varied based on testing rationale. SARS-CoV-2 testing strategies should bear in mind that primary decision makers seem more accepting of uncomfortable sampling techniques when they see benefit to testing, and perceptions of test accuracy can supersede the desire for more comfortable sampling. Patient and family preferences should be considered by public health policy makers, especially when testing modalities are comparable. Further study of SARS-CoV-2 sampling accuracy and acceptability will help determine the ideal application for each modality.

## Supporting information

**S1 Checklist. STROBE statement—Checklist of items that should be included in reports of cross-sectional studies.**
(DOCX)

**S1 File. Supporting information–Survey.**
(DOCX)

## Author Contributions

**Conceptualization:** François Gagnon, Maala Bhatt, Roger Zemek, Richard J. Webster, Stephanie Johnson-Obaseki, Stuart Harman.

**Data curation:** François Gagnon, Stephanie Johnson-Obaseki, Stuart Harman.

**Formal analysis:** François Gagnon, Maala Bhatt, Richard J. Webster.

**Investigation:** François Gagnon, Stephanie Johnson-Obaseki.

**Methodology:** François Gagnon, Maala Bhatt, Roger Zemek, Richard J. Webster, Stephanie Johnson-Obaseki, Stuart Harman.

**Resources:** Maala Bhatt, Roger Zemek, Stephanie Johnson-Obaseki, Stuart Harman.

**Supervision:** Maala Bhatt, Roger Zemek, Stuart Harman.

**Writing – original draft:** François Gagnon.

**Writing – review & editing:** Maala Bhatt, Roger Zemek, Richard J. Webster, Stephanie Johnson-Obaseki, Stuart Harman.

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
