## [Decision Letter · Decision Letter 0]

8 Apr 2022

PONE-D-22-07725Nasopharyngeal Swabs vs. Saliva-based Sampling for SARs-CoV-2 Detection: A Cross-Sectional Survey of Acceptability for Caregivers and Children after Experiencing Both MethodsPLOS ONE

Dear Dr. Gagnon,

Thank you for submitting your manuscript to PLOS ONE. After careful consideration, we feel that it has merit but does not fully meet PLOS ONE’s publication criteria as it currently stands. Therefore, we invite you to submit a revised version of the manuscript that addresses the points raised during the review process.

ACADEMIC EDITOR: As appended below, the reviewers have raised major concerns/critiques (reviewer # 1 is against publication) and suggested further justification/work to consolidate the findings. Do go through the comments and amend the MS accordingly

We look forward to receiving your revised manuscript.

Kind regards,

A. M. Abd El-Aty

Academic Editor

PLOS ONE

Journal Requirements:

Reviewers' comments:

Reviewer's Responses to Questions

**Comments to the Author**

1. Is the manuscript technically sound, and do the data support the conclusions?

Reviewer #1: Yes

Reviewer #2: Partly

Reviewer #3: Yes

Reviewer #4: Yes

2. Has the statistical analysis been performed appropriately and rigorously? 

Reviewer #1: I Don't Know

Reviewer #2: Yes

Reviewer #3: Yes

Reviewer #4: Yes

3. Have the authors made all data underlying the findings in their manuscript fully available?

Reviewer #1: Yes

Reviewer #2: Yes

Reviewer #3: Yes

Reviewer #4: Yes

4. Is the manuscript presented in an intelligible fashion and written in standard English?

Reviewer #1: Yes

Reviewer #2: Yes

Reviewer #3: Yes

Reviewer #4: Yes

5. Review Comments to the Author

Reviewer #1: The authors present a very relevant study, one which can be informative for future diagnostic practices. We have seen first-hand through the pandemic how testing aversion can lead to delayed testing and testing avoidance, both which can have serious consequences for public health. We must learn from our experiences through the pandemic and continue to innovate the rest of our clinical diagnostics to improve acceptability to the patient to encourage greater adherence to testing - and less fear of clinical space. While a few questionnaires have been issued to evaluate people's thoughts and experiences with sampling technique, this is the first that I am aware of to also study the physical response to these samples. Thus, I commend the authors on their study and bringing this issue to light.

Major comments:

Lines 74-76: There was a recent study evaluating test acceptability in schools which may require this text to be updated a little: https://journals.sagepub.com/doi/full/10.1177/00333549221074395. The authors might like to review this recent publication in case any of the references need to be updated or if this paper adds any additional support to their work: https://www.dovepress.com/acceptability-of-community-saliva-testing-in-controlling-the-covid-19--peer-reviewed-fulltext-article-PPA

Methods: the sampling procedure for both NP and saliva sampling is missing. This is important to the study for the reader to understand especially, which saliva sampling technique they received and how they found this. It would also be helpful to attach the saliva sampling SOP/IFU into the appendix so people can evaluate this - ie, were people coaxed? This can help with collection. The authors could also discuss that 1 mL is a relatively large amount to collect considering many major saliva tests only require 50-300 ul of sample, meaning that less could be collected, further improving acceptability.

Conclusion: why is there a sudden mention of Mers?

Minor comments:

As 'NP' and 'nasopharyngeal' contribute the same word count and as 'saliva-based' and 'SB' also contribute the same word count (or in many places could simply be 'saliva'), I recommend that 'saliva' and 'nasopharyngeal' be written in full. As this work will have broad appeal, this will be easier for the less-experienced reader to follow along with.

Update 'SARs-CoV-2' to 'SARS-CoV-2' throughout.

Line 225: This makes a good reference for how it is the robustness of saliva testing (and sample collection) that can affect sensitivity (rather than straight 'saliva' itself) - this is important as with robust methods, saliva performs equal to or better than swabs: https://www.thelancet.com/journals/lanres/article/PIIS2213-2600%2821%2900178-8/fulltext

Reviewer #2: Major Comments

- Why were there two different saliva sampling methods used in the study? As this is evaluating preference the NP stays constant, but a sponge collection and spit would add additional variables to the already limited number of patients. Ideally additional participants would be added to remove the sponge method. If not possible, the data should be separated between method types.

- I would suggest adding more details about the questions used in the survey. At the moment the material and methods of what was asked was unclear and requires going to supplemental data to review the survey to understand what was asked.

Minor Comments

Ln 257 and 261: This seems to be a typo as its stated as Mers and not SARS.

One complaint and issue with saliva testing is obtaining enough volume to perform all testing. Was there any issues to obtain the 1mL from subjects?

I would add in the limitations that the majority of primary decision makers had some college education or higher, which may have skewed results, especially when asking about sensitivity.

Reviewer #3: This was a very nice study looking at the acceptability of saliva and NP swab for testing children for COVID-19. This is particularly interesting because almost all other studies have focused solely on measures of test sensitivity and accuracy as opposed to patient comfort and ease. While NP swabs are the gold-standard, they are onerous to perform and so it is good to know when the slight increase in sensitivity for the test might outweigh concerns about the patient anxiety and comfort, particularly children. I thought the paper was quite good overall and suggest just some additional discussion/background to help readers place the results in context of other studies on COVID-19 testing:

1. I think it might be helpful to include some discussion of limitations of both techniques in general. For instance some patients cannot do a saliva test because they cannot self collect a sample (e.g., very young infants, patients with certain disabilities). Some patients are very difficult to get NP swabs on because of nasal anatomical/pathological issues. There's also typically a difference in overall cost of the tests with saliva typically being a bit cheaper overall since saliva tests do not require specialized swabs/supplies or highly trained medical staff to collect.

2. The authors also do not discuss other less invasive swabs (anterior nasal, oral, etc). There have been some studies using these in other settings where NP swabs are not easy to administer.

3. One of the more interesting findings is that parents would still be ok using saliva over NP swabs even if the NP swab was 10% more accurate. This might be a good place to discuss the relative accuracy of these two testing modalities.

4. This was done on a population of patients that had symptoms or known exposures to SARS-CoV-2 and thus, one would assume that they would be more likely to want better diagnostics than they would if this test was for a screening application. It may be useful to have a follow up survey of healthcare professionals or reference some work in the literature that surveys clinicians to compare the acceptability of saliva over NP swabbing in different applications. I would assume that clinicians would in general prefer NP swabbing for diagnostics as it is the gold standard but the data on pain and patient acceptability may sway their decision based on the reason for the COVID-19 test.

Overall, this was a well designed study, which fills a clear gap in our understanding of the COVID-19 testing landscape.

Reviewer #4: This is a very relevant, well-conceived and well-written study of patient preferences in the context of a rapid increase in the experience of upper respiratory sampling in all age groups. I listened to my asymptomatic niece scream during a required NP swab for a dental appointment, with an absolutely massive swab, and wondered why we couldn’t have SB screening for kids like in BC?? How individuals weigh the risks and benefits in the context of sampling quality is a fascinating element of this study – that more pain is acceptable with more gain, and whether the case is symptomatic, asymptomatic, and/or required to test might all influence which test is preferable. I am very intrigued by the older kids who preferred NP, such an interesting perspective.

Your opening statement in the introduction seems no longer quite accurate post-Omicron. Not suggesting any specific changes, but it would be nice to see some additional perspectives in Intro/Discussion on how your findings might be different now, when the “scenario” is declining availability of free, trackable testing and rampant pauci-symptomatic infections at school?

How did you put the survey on the electronic device and link to RedCap, in detail? Software? Scripts? Device used? This is just for my information because I want to do this next month!

I am curious about differences in swabs used for children (thickness, fluffiness vs. sharpness of swab head) and how painful each type might be, for example, if an “adult-sized” swab was used on a little one – could this account for some of the difference between yours and Moisset’s study?

Figure 1: Add y axis title (“% of participants”), remove “%” from tick labels, y axis should end at 100, not go past it. In legend, “Saliva test” should be “Saliva-based test” or just use “SB” and “NP” to be consistent with text. Figure Caption: “at least likely” is ambiguous, do you mean just the top two Likert points? Always use curly quotation marks.

Figure 2: Add y axis title (“% of participants”), remove “%” from tick labels, only label every second tick (at 10% rather than 5% intervals). Figure caption is unclear, how about: “Self-reported pain after NP swab using revised Face Pain Scale (FPS-Re, percent of participants).”

Figure 3: In legend, “Saliva test” should be “Saliva-based test” to be consistent with text. Figure caption is unclear, how about: “Preferred test in case of future need for testing (percent of participants).”

Is “Interpretation-Limitations-Conclusion” a suggested PLoS One structure? Just “Discussion” (no sub-heads) seems preferable.

This paper is low on refs. It is a unique study, so this makes sense, however some additional context about saliva testing for COVID and other infections, and pediatric screening vs. diagnostics in general, would be much appreciated.

Supporting Information/"Appendix": Not sure if I see the utility of attaching these. STROBE is not referenced anywhere else that I noticed, so what is it supporting? The survey “appendix” seems a bit repetitive to the main text and is only referenced once (reference should be to SI, not Appendices).

Minor comments and style suggestions (please take or leave):

Throughout – Ref numbers and Figure refs on the wrong side of the punctuation or with missing spaces between ref and text.

Title - Line 1: typo in SARS-CoV-2

Line 46: “prior” instead of “a priori”

Line 62: “factors that influence sampling” instead of “factors sampling”

Line 66 and 68: parentheses instead of square brackets

Line 70: hyphen in “under-represented”

Line 72: delete “It has been shown that”

Line 95: “decision” instead of “decisions” – Just curious, when is assent appropriate?

Line 110: Adolescents – what is exact age group?

Line 113: “our own” instead of “a de novo”

Line 118: italicize e.g.

Line 119: Too much punctuation - “likely, see Appendix)” instead of “likely; (see Appendix))

Line 120: Square brackets look weird to me in paragraph text.

Line 121: “by asking yes or no” instead of “with direct binary”, “concluding” instead of “final”

Line 129-31: A bit wordy… “were dichotomized to likely and not likely (including neutral).”

Line 142: Wordy… “Convenience sample size was 50 participants”.

Table 1 – Needs cleaning up, too many ugly square brackets! Spaces after commas. No need for “+” or “[%]” or “, yes” – Remove extra space before “CEGEP”.

Line 174 and 176: Space before “ 2”. Space before “ 3”.

Line 182: Period after “vs.”

Line 182-4: Sounds like the parent wants the numbing spray for themselves – “prefer a nasal numbing spray be offered to their child” ?

Line 193: “acceptability” instead of “acceptably”

Line 195: “contact of” instead of “contact with”, comma after “case,”

Line 203: “et al.” instead of “et. al.”

Line 212: “not testing is less obvious” instead of “not testing is not as obvious”

Line 215: “testing is perceived as less valuable” instead of “the value of testing is not seen as high”, italicize e.g.

Line 219: “most effectively” instead of “best”

Line 225: “likely varies in different” instead of “is likely different in different”

Line 226-7: italicize “et al.”

Line 227: “that sensitivity” instead of “that te sensitivity”

Line 229: “ranged from 22-94%” instead of “to be as low as 21.9% and as high as 93.9%”

Line 229-30: “comparator” instead of “gold standard”

Line 231: no hyphen in “ongoing”

Line 238: no hyphen in “inpatient”

Line 239: “understanding feasibility and acceptability of saliva-based screening” instead of “understanding acceptance”

Line 241: “since” instead of “as”

Line 242-3: “designed a survey to assess our primary outcome, based on best practices (6).”

Line 247: Briefly explain what is a “collider”?

Line 248: “youth and their families” instead of just “youth”, “is valuable regardless” instead of “justifies the risk”

Line 249-51: Axe the last two sentences of this paragraph.

Line 256: “varied” instead of “is more prone to vary”

Line 257: “SARS” instead of “Mers”

Line 259: “supercede” is mis-spelled

Line 261: “comparable” instead of “non-inferior or equivalent”, “SARS” instead of “Mers”

Line 262: “ideal application for each modality” instead of “ideal use for the differing modalities”.

Line 269-74: Add cédille to “François”

6. PLOS authors have the option to publish the peer review history of their article (what does this mean?). If published, this will include your full peer review and any attached files.

Reviewer #1: No

Reviewer #2: No

Reviewer #3: No

Reviewer #4: **Yes: **John J. Schellenberg

---

## [Author Response · Author response to Decision Letter 0]

2 Jun 2022

Dear reviewers,

We are truly thankful for your time and efforts invested in reviewing our manuscript. Your insightful comments helped us improve the paper significantly. 

We have addressed comments and suggestions on a point-by-point basis in the "Reviewers response" document. We invite you to review this document as well as the new version of the manuscript.

We are hopeful that you will find our modifications to your satisfaction and welcome questions, comments and suggestions. 

Sincerely,

Francois Gagnon, MD, FRCPC

Pediatric Emergency Medicine Fellow

Children's Hospital of Eastern Ontario

---

## [Decision Letter · Decision Letter 1]

21 Jun 2022

Nasopharyngeal Swabs vs. Saliva Sampling for SARS-CoV-2 Detection: A Cross-Sectional Survey of Acceptability for Caregivers and Children after Experiencing Both Methods

PONE-D-22-07725R1

Dear Dr. Gagnon,

We’re pleased to inform you that your manuscript has been judged scientifically suitable for publication and will be formally accepted for publication once it meets all outstanding technical requirements.

Kind regards,

A. M. Abd El-Aty

Academic Editor

PLOS ONE

Additional Editor Comments (optional):

Please add the product name/supplier details of sponge-based saliva kit in the galley proof, as requested by reviewer # 1

Reviewers' comments:

Reviewer's Responses to Questions

**Comments to the Author**

1. If the authors have adequately addressed your comments raised in a previous round of review and you feel that this manuscript is now acceptable for publication, you may indicate that here to bypass the “Comments to the Author” section, enter your conflict of interest statement in the “Confidential to Editor” section, and submit your "Accept" recommendation.

Reviewer #1: (No Response)

Reviewer #2: All comments have been addressed

Reviewer #3: All comments have been addressed

2. Is the manuscript technically sound, and do the data support the conclusions?

Reviewer #1: Yes

Reviewer #2: Yes

Reviewer #3: Yes

3. Has the statistical analysis been performed appropriately and rigorously? 

Reviewer #1: I Don't Know

Reviewer #2: Yes

Reviewer #3: Yes

4. Have the authors made all data underlying the findings in their manuscript fully available?

Reviewer #1: Yes

Reviewer #2: Yes

Reviewer #3: (No Response)

5. Is the manuscript presented in an intelligible fashion and written in standard English?

Reviewer #1: Yes

Reviewer #2: Yes

Reviewer #3: Yes

6. Review Comments to the Author

Reviewer #1: The authors have submitted a robust revision of their work. The only minor comment I have is that the sponge-based saliva kit used needs to have product name/supplier details.

Reviewer #2: The authors addressed all of my comments that I presented on the first review. No further changes are needed after the revisions.

Reviewer #3: The authors have addressed all my prior concerns. The paper is well written, and relevant to the current ongoing pandemic situation.

7. PLOS authors have the option to publish the peer review history of their article (what does this mean?). If published, this will include your full peer review and any attached files.

Reviewer #1: No

Reviewer #2: No

Reviewer #3: No

---

## [Editor Report · Acceptance letter]

30 Jun 2022

PONE-D-22-07725R1 

Nasopharyngeal Swabs vs. Saliva Sampling for SARS-CoV-2 Detection: A Cross-Sectional Survey of Acceptability for Caregivers and Children after Experiencing Both Methods 

Dear Dr. Gagnon:

I'm pleased to inform you that your manuscript has been deemed suitable for publication in PLOS ONE. Congratulations! Your manuscript is now with our production department. 

Kind regards, 

on behalf of

Prof. A. M. Abd El-Aty 

Academic Editor

PLOS ONE